# Nomilin and Its Analogues in Citrus Fruits: A Review of Its Health Promotion Effects and Potential Application in Medicine

**DOI:** 10.3390/molecules28010269

**Published:** 2022-12-29

**Authors:** Zhenyu Zhou, Yingxuan Yan, Hongli Li, Yaru Feng, Cheng Huang, Shengjie Fan

**Affiliations:** School of Pharmacy, Shanghai University of Traditional Chinese Medicine, Shanghai 201203, China

**Keywords:** nomilin, citrus fruits, tetranortriterpenoid, biological activities, health promotion

## Abstract

Nomilin is one of the major limonoids, which are plant secondary metabolites also known as tetranortriterpenoids. Nomilin is found mostly in common edible citrus fruits including lemons, limes, oranges, grapefruits, mandarins, along with traditional Chinese medicines derived from citrus fruits, such as tangerine seed, tangerine peel, fructus aurantii immaturus, etc. A number of studies have demonstrated that nomilin and its analogues exhibit a variety of biological and pharmacological activities. These include anti-cancer, immune-modulatory, anti-inflammatory, anti-obesity, anti-viral, anti-osteoclastogenic, anti-oxidant, and neuro-protective effects. Thus, nomilin and its analogues have emerged as a potential therapy for human diseases. The purpose of this review is to chronicle the evolution of nomilin research from examining its history, structure, occurrence, to its pharmacological and disease-preventing properties as well as its potential utilization in medicine and food science.

## 1. Introduction

Citrus fruits are one of the largest crops produced globally with grapefruits, lemons, oranges, and tangerines being among the most important. *Citrus* species possess a wide variety of bioactive compounds which can promote human health, e.g., vitamin C, flavonoids, limonoids, folic acid, dietary fiber, potassium, etc. [1]. However, the limonoids, a family of tetranortriterpenoids with an attaching furan ring, are unique to fruit [2]. Limonoids can be present as water-soluble “tasteless” glucosides and as water-insoluble “bitter” aglycones [3]. Limonin-17 β-d-glucopyranoside (Figure 1) is the most common glucoside, while limonin (Figure 1) and nomilin (Figure 1) are the most abundant aglycones, and both of them are responsible for the bitterness of citrus fruits and juices [4]. Results of sensory analysis indicated a bitter taste threshold of 6 ppm [5] for limonin, and 6 ppm [6] or 3 ppm [7] for nomilin. Nevertheless, the contribution of nomilin to the bitter taste of juices is minor; it occurs mainly in grapefruit juice [8]. During the past few years, the compound nomilin was proved to possess a number of biological activities, including anti-cancer [9,10,11,12,13], immune-modulatory [14], anti-inflammatory [15], anti-obesity [16], anti-viral [17], anti-osteoclastogenic [18], and neuro-protective effects [19]. Hence, this natural limonoid has been extensively investigated.

Although several review articles cover the properties of citrus limonoids [1,20,21,22], so far, to the best of our knowledge, there is no specialized review about the major limonoid nomilin. Therefore, the present review aimed to summarize the literatures about nomilin, focusing on its history, chemical structure, occurrence, along with the important research findings regarding its pharmacological activities and protective effects against diverse diseases.

## 2. Historical Perspective

Citrus is one of the major fruit crops grown worldwide. Up to date, the production of citrus fruits is located in five continents. It is considered that citrus is native to the subtropical and tropical areas of Southeast Asia including China, India, and the Malay Archipelago [23,24,25,26,27], but the exact place of origin of citrus fruits is still under controversy. According to the Chinese ancient documents, the earliest record that refers to citrus appeared around 2205 to 2197 BC in the Xia Dynasty [28,29], whereas recent research suggests that some commercial citrus species, e.g., mandarins, lemons, oranges, originated in Southeast Asia; the true origins of citrus fruit are Australia, New Caledonia, and New Guinea [30]. The spread of citrus to America was achieved by Spanish and Portuguese settlers, and the citrus first appeared in Florida and California around 1655 and 1769, respectively [31]. Commercial production of citrus started only in the latter half of the 19th century. Thereafter, the production, processing, and global trade of citrus expanded vastly and made citrus as the most important fruit in the world [26].

In the *citrus* genus, the limonoids have been gaining attention due to their contribution to the bitterness of citrus fruits and juices [32,33], along with their various biological activities, especially anticancer activity [34]. Large amounts of limonoids are present in the processing waste of citrus fruits, including seeds, peels, and molasses; hence, mass production of limonoids can be economically profitable [32,35]. The first well-recognized bitter principle limonin was isolated by Bernay in 1841 and was obtained from a series of citrus and related *Rutaceae* [36]. From the mother liquors of limonin from the seeds of lemons and oranges, a second bitter principle was derived by Emerson in 1948 [37]. Since this substance is closely associated with limonin and similar in chemical properties to it, the name “nomilin” is proposed [37].

## 3. General Characterization of Nomilin

### 3.1. Structure and Chemistry

Citrus limonoids are comprised of two nucleus structures. The first general nucleus consists of five rings, while the second structure is composed of four rings designated as A, B, C, and D [38]. The structure of nomilin exemplifies the second general nucleus (Figure 1). Further, nomilin of the formula C_28_H_34_O_9_ contains a dilactone structure and a furan ring attached to the closed D-ring at C-17.

Nomilin can remain in solution in ethanol containing methylene chloride and dissolved in hot 2-propanol; therefore, it is easily separated from limonin [37]. In addition, nomilin has a tendency to form solvates with many solvents, which causes the appearance of crystals from diverse solvents to be significantly different. As nomilin has only one active hydrogen, it is difficult to acetylate by refluxing with acetic anhydride and sodium acetate or with acetic anhydride and pyridine for several hours [37]. Except under the strong activating influence of pyridine, nomilin cannot react with carbonyl reagents. In comparison with the beautifully crystalline oxime of limonin, the oxime, semicarbazone, and 4-phenylsemicarbazone of nomilin are amorphous. Owing to the decomposition of these substances, their melting points are indefinite [37]. Moreover, nomilin has a hydroxyl group, which can react with methylmagnesium iodide but not with acetic anhydride [37].

### 3.2. Natural Plant Extraction and Biosynthesis

#### 3.2.1. Natural Plant Extraction

Pomelo: Hu et al. [39] carried out a study on the extraction of nomilin from the segment membrane of Guanxi pomelo (origin: Fujian Province, China). After the segment membrane was completely degreased with petroleum ether, 20 times (*v*/*m*) acetone was added, and the mixture with pH 7.0 was heated in a water bath for 15 h at 40 ℃. The yield of nomilin was 0.033%. Sun et al. [40] used the water bath heating method to obtain nomilin from the segment membrane of Zhoushan pomelo (origin: Zhejiang Province, China). In this method, the solvent extract was heated in a water bath for 60 min at 50 ℃, then identified and determined by Thin Layer Chromatography (TLC) and High-Performance Liquid Chromatography (HPLC). The recovery of nomilin was 91.1%.

Citrus: A study was conducted to isolate nomilin from citrus seed oil by Emerson [41] in 1951. The citrus seed oil was firstly mixed with petroleum ether. After filtration, some bitter substances were separated and extracted with acetone. The petroleum ether solution was then extracted three times with 95% methanol. After that, the acetone and methanol extracts were further concentrated. By using this process, 80 g of nomilin was obtained from 30 gallons of citrus seed oil. Furthermore, in order to optimize nomilin yield at different levels of supercritical carbon dioxide extraction parameters, including extraction pressure, extraction temperature, and dosage of entrainer (95% ethanol), Luo et al. [42] analyzed the process conditions for supercritical carbon dioxide extraction of nomilin from citrus seeds. The results demonstrated that the optimal levels of the above-mentioned extraction parameters for achieving a maximum nomilin yield (0.63%) were as follows: extraction pressure 25 MPa, extraction temperature 51 ℃, and entrainer dosage 1.5 mL/g.

Sweet orange: The extraction of nomilin from sweet orange seeds was studied by Tian et al. in 1999 [43]. After degreasing with petroleum ether, sweet orange seeds were extracted with acetone. The extracted solution was concentrated under reduced pressure, then dissolved in dichloromethane and filtered. Two times the volume of isopropanol was added into the filtered solution. This mixture was then placed in the refrigerator at 4 ℃. After crystallization, the mother liquor was evaporated and separated by silica gel column chromatography (eluent: ethyl acetate-methanol). The yield of nomilin was 0.04%.

#### 3.2.2. Biosynthesis and Chemical Synthesis

It is considered that deacetylnomilinic acid is the most likely initial precursor for the biosynthesis of citrus (*Rutaceae*) limonoids [44]. Deacetylnomilinic acid may be biosynthesized from acetate, mevalonate (MVA), and farnesyl pyrophosphate in the phloem region of stems and can be transformed to nomilin [45]. Both deacetylnomilinic acid and nomilin can be translocated from stems to other plant parts such as fruit tissues, seeds, and leaves [44].

Deacetylnomilin was isolated from the seeds of the citrus species and *poncirus trifoliate* by Dreyer in 1964 [46]. The method for the chemical synthesis of nomilin is as follows [46]: Deacetylnomilin is firstly acetylated by refluxing with acetic anhydride-pyridine (Figure 2). The mixture is then decomposed with water and extracted with chloroform. After removal of the solvent, nomilin is obtained.

### 3.3. Bioavailability and Absorption

In a study conducted by Cai et al. [47], the pharmacokinetic parameters of nomilin in rats were investigated. The results showed that the concentration of nomilin in blood increased rapidly and reached the peak concentration at 1.7 h after nomilin was orally (50 mg/kg) administrated to rats. By comparison, the blood concentration of nomilin decreased rapidly and then reduced gradually after nomilin was intravenously administrated. The oral bioavailability of nomilin was 4.2%. This result suggests that the blood content of nomilin is low after it was orally administrated. The partition coefficient (Po/w) can be used to determine the distribution of drugs within the body. Since nomilin is a hydrophobic substance, and its partition coefficient value is 2.15, it may be poorly absorbed and permeated [47]. Although the bioavailability of nomilin is very low due to its low solubility and poor absorption, its bioavailability could be improved by a nanoparticle-based delivery system [48]. Nomilin nanoparticles prepared using a poly (lactic-co-glycolic acid) with didodecyl dimethyl ammonium bromide surfactant were shown to have the mean particle sizes 135–188 nm, and exhibited a slow release pattern and significant inhibition of a-amylase (72% to 29.9% of pure nomilin) and angiotensin-converting enzyme activity (100% to about 23% of pure nomilin), indicating that the solubility and bioavailability of nomilin could be improved by a nanoparticle-based delivery system [48].

## 4. Variation in Nomilin Content in the Fruits of Several Citrus Varieties

In a recent study, Huang et al. [49] accessed the nomilin content in the fruits of eight citrus varieties, including Ehime No. 34, Ehime No. 38, lemon, Navelina Navel, Orah, Tarocco No. 4, Tarocco No. 8, and Yellow Tacopon by HPLC at different fruit growth and maturation stages. The results (Figure 3) demonstrated that Yellow Tacopon had the highest content of nomilin at the stages of young fruit and fruit falling, while Orah presented the lowest nomilin content at the same stages. Nevertheless, Orah showed the highest content of nomilin, followed by lemon, and Tarocco No. 4 presented the lowest nomilin content at the fruit expanding stage. In all tested varieties, the content of nomilin could not be detected at the mature stage.

As shown in Figure 3, the content of nomilin in Ehime No. 34, Ehime No. 38, lemon, Navelina Navel, Tarocco No. 4, and Yellow Tacopon showed an upward trend between the fruit falling and young fruit stages, and decreased after the young fruit stage until it could not be detected at the mature stage. In contrast, Orah and Tarocco No. 8 presented the highest nomilin content at the fruit expanding stage and fruit falling stage, respectively, and then decreased gradually. Overall, these results indicated that the maximum nomilin content presented at the young fruit stage in most tested citrus varieties.

## 5. Bioactivities and Health Beneficial Effects of Nomilin

In the past decades, the health benefits of nomilin on animals and humans were the subject of numerous research studies. Strong and compelling evidence suggests that nomilin has the capacity to inhibit the growth and proliferation of many types of cancers [9,10,11,12,13], act as an anti-inflammatory agent [15], and modulate the severity of human immunodeficiency virus infections [17]. Additional actions include suppressing diet-induced obesity and hyperglycemia [16], enhancing the elimination of toxic electrophiles by phase II detoxifying enzymes such as glutathione-*S*-transferase (GST) [50,51].

### 5.1. The Detoxification Effects of Nomilin

Nomilin was shown to be a potent inducer of phase II drug metabolism and detoxifying enzymes’ GST activity in the liver, the small intestinal mucosa, and the forestomach, but not in the lung and colon in mice [50]. Meanwhile, other limonoids, such as obacunone, isoobacunoic acid, and ichangin, were only shown to be enzyme inducers in the liver and some other tissues studied [50]. Nomilin not only induced GST activity against 1-chloro-2,4-dinitrobenzene in the stomach, the intestine, and the liver in mice, but also increased the activity of another Phase II detoxifying enzyme NAD(P)H: quinone reductase, which protects against cytotoxicity by catalyzing the reduction of quinones, in mice [52]. The results suggest that nomilin may provide a protective effect against the onset of various cancers in the liver and the small intestine by inducing the activity of certain phase II detoxifying enzymes.

Nomilin may increase phase II enzyme activity, but not phase I drug and toxin metabolism enzyme cytochrome P450, in the liver and the small intestine of rats [53]. In contrast, nomilin was reported to inhibit cytochrome P450 3A4 (CYP3A4) activity, which inhibited CYP3A4 activity in a time-, concentration-, and NADPH-dependent manners [54,55]. A recent study also showed that nomilin inhibits CYP3A4 and Daunorubicin transport in Caco-2 and calcein-AM uptake in LLC-PK1 and LLC-GA5-COL300 in the concentration-dependent manner, indicating that nomilin could activate both CYP3A4 and P-gp [53]. Taken together, nomilin is possibly used as a bioavailability modulator. However, it also raised the concerns about the concomitant use of nomilin and other drugs, which may cause clinical safety problem.

### 5.2. Anti-Oxidant

Using free radical 1,1-diphenyl-2-picrylhydrazyl analysis, the antioxidant activity of nomilin (mixed with limonin) was identified, although it is lower than that of synthetic antioxidant butylated hydroxyanisole [56]. A food technology study showed that the amount of nomilin and limonin correlated well with the total antioxidant activity, and nomilin was one of the major contributors to the antioxidant activity of the product [57]. The antioxidant capacities of nomilin were 2.9–8.3 times higher than that of vitamin C using a beta-carotene bleaching assay [58]. However, a study showed that the antioxidant effect of nomilin is equivalent to those of negative control cinnamic acid, indicating that it does not possess an inherent antioxidant capacity and should not be considered antioxidants [59]. The discrepancy between these reports needs to be further studied.

### 5.3. Anti-Cancer Activity

Several in vitro and in vivo studies revealed that nomilin exhibits the protective effects against certain types of cancer. Nomilin has been shown to inhibit the proliferation of estrogen receptor-negative (ER-) and -positive (ER+) human breast cancer cells in culture [60], and the anti-proliferative activity of nomilin is mediated via caspase-7 dependent pathways in breast cancer cells [13]. Tian et al. [10] studied the effect of nomilin against a series of human cancer cell lines including the breast (MCF-7), cervix (HeLa), leukemia (HL-60), liver (Hep G2), ovary (SKOV-3), and stomach (NCI-SNU-1). Data from this study showed that the growth-inhibitory effect of nomilin against MCF-7 cells was marked, and its antiproliferative activity was time- and dose-dependent. With use of flow cytometry, it was also found that nomilin could induce apoptosis in MCF-7 cells. Nomilin inhibited pancreatic cancer (Panc-28) cells’ proliferation and induced apoptosis through the cleavage of caspase-3, and decreased the mitochondrial membrane potential and expression of apoptosis-related proteins [61]. Nomilin also exhibited toxic effects against human SH-SY5Y neuroblastoma and Caco-2 colonic adenocarcinoma cells [3]. These results suggest that nomilin may exert a multifaceted lethal action against cancer cells in several tissues.

In a study conducted by Lam et al. [9], nomilin was found to suppress benzo(α)pyrene (BP)-induced neoplasia in the forestomach of ICR/Ha mice. At a 10 mg dose of nomilin, the number of mice with tumors was decreased from 100% to 72%, and the number of tumors in each mouse was significantly reduced. In addition, nomilin, when given three doses (5 and 10 mg/dose/animal) every two days, induced increased GST activity 3.00 and 4.17 times above that of controls, respectively, in the small intestinal mucosa of test mice, while the increases in GST activity in the liver of test mice were 2.48 and 3.44 times over the control, respectively.

Pathological angiogenesis is a critical factor in cancer development. It increases the risk of fatal metastases. Thus, Pratheeshkumar et al. [11] investigated the antiangiogenic activity of nomilin. The results showed that the tumor-directed capillary formation was significantly inhibited by the treatment with nomilin. Furthermore, nomilin markedly reduced serum proinflammatory cytokines including IL-1β, IL-6, and tumor necrosis factor (TNF)-α along with serum NO and vascular endothelial growth factor (VEGF) levels, whereas the antiangiogenic factors IL-2 and tissue inhibitor of metalloproteinase (TIMP)-1 were significantly increased. Studies using human umbilical vein endothelial cells demonstrated that administration of nomilin remarkably hindered endothelial cell proliferation, migration, invasion, and tube formation. In vitro studies indicated that nomilin significantly inhibited microvessel sprouting. These data suggest that nomilin may be a potential angiogenic inhibitor by downregulating the activation of matrix metalloproteases (MMPs), production of VEGF, NO, and proinflammatory cytokines as well as upregulating IL-2 and TIMP.

In another study by Pratheeshkumar et al. [12], the antimetastatic potential of nomilin was researched. It was found that treatment with nomilin inhibited metastatic lung tumor nodule formation and significantly increased the survival rate of the tumor-bearing animals. Nomilin exhibited the inhibition of tumor cell invasion and activation of MMPs. Moreover, nomilin treatment induced a downregulated Bcl-2 and cyclin-D1 expression and upregulated p21, p27, p53, Bax, caspase-3, and caspase-9 gene expression in B16F-10 cells. This study also revealed that the activation and nuclear translocation of antiapoptotic transcription factors such as nuclear factor (NF)-κB, cAMP responsive element-binding protein, and ATF-2 in B16F-10 cells could be suppressed by nomilin.

A structure-function relationship study demonstrated that nomilin containing the different functional groups had differential effects on the p38 MAP kinase activity in human aortic smooth muscle cells. Nomilin displayed the highest (38%) inhibition of p38 MAP kinase activity compared to limonin (19%), deacetyl nomilin (19%), and defuran nomilin (17%). Furthermore, TNF-α induced p38 MAP kinase activity in the smooth muscle cells was completely inhibited by nomilin, suggesting that nomilin is the potent natural inhibitor for p38 MAP kinase activity in human aortic smooth muscle cells. These data also suggest that a seven-membered A ring with an acetoxy group of nomilin seems to be essential for its inhibitory activity on p38 MAP kinase [62].

### 5.4. Immunomodulation

An in vivo study [14] demonstrated that intraperitoneal treatments with five doses of nomilin could enhance the total white blood cells account, and the maximum count was observed on the sixth day in mice. In addition, the bone marrow cellularity and α-esterase positive cells were also increased by nomilin. Nomilin also enhanced the antibody titer and the number of plaque forming cells in the spleen and significantly inhibited the delayed type hypersensitivity reaction. These results indicate that nomilin has immunomodulatory activity.

### 5.5. Anti-Inflammation

The inflammatory cytokines play a pathophysiological role in various diseases. Kim et al. [15] provided the evidence that treatment with nomilin could significantly inhibit TNF-α-induced proliferation in human aortic smooth muscle cells (HASMCs). Additionally, nomilin markedly reduced the phosphorylation levels of the inhibitor of NF-κB (IκBα) and suppressed the inflammatory reaction in TNF-α-treated HASMCs. These results indicated that the anti-proliferative activity of nomilin on TNF-α-induced HASMCs results from apoptosis through inhibition of NF-κB mediated inflammatory signaling, which may be beneficial to atherosclerosis. Nomilin also significantly downregulated the pro-inflammatory cytokines interleukin-1β, IL-6, IL-8, TNF-α, and NF-κB in lipopolysaccharide-stimulated mouse RAW 264.7 macrophages and HT-29 human colon epithelial cells and showed efficacy against inflammatory bowel disease [63].

### 5.6. Anti-Obesity Effect

TGR5 (G protein-coupled bile acid receptor 1, M-Bar), a member of the G protein-coupled receptor family, may promote energy expenditure and improve glucose homeostasis and is a key target in treating metabolic diseases. A study conducted by Ono et al. [16] found that nomilin is a novel TGR5 agonist. It exerts anti-obesity and anti-hyperglycemic effects in high-fat diet (HFD)-fed mice. When obese mice fed a HFD either alone or in combination with 0.2% *w/w* nomilin for 77 days, nomilin-treated mice had lower body weight, serum glucose, serum insulin, and increased glucose tolerance compared with those fed a nomilin-free HFD. The data obtained in this study suggest a novel biological function of nomilin as a potential anti-obesity and anti-hyperglycemic agent that acts likely through the activation of TGR5. Further study showed that human TGR5 has higher nomilin responsiveness than mouse TGR5. Using mouse-human chimeric TGR5 revealed that three amino acid residues (Q77(ECL1), R80(ECL1), and Y89(3.29)) are important in the human TGR5-nomilin interaction, and demonstrated that four hydrophilic hydrogen-bonding interactions occur between nomilin and human TGR5 [64], which provided the structure basis for the modification of nomilin as anti-obesity drugs.

### 5.7. Anti-Viral

It was reported by Battinelli et al. [17] that nomilin was effective in inhibiting human immunodeficiency virus (HIV)-1 replication in all cellular systems used, including in vitro infected human peripheral blood mononuclear cells (PBMC), naturally infected PBMC, and in vitro infected monocytes-derived macrophages (M/M), which is one of the most important viral reservoirs in vivo and a key target for a successful therapy. The mechanism of the anti-viral action of nomilin is considered to suppress in vitro HIV-1 protease activity. Recently, nomilin was reported to inhibit SARS-CoV-2 virucidal activity effectively in Vero E6 cells in a dose-dependent manner, with IC50 about µg/mL. The effects may be achieved through a dual virucidal–antioxidant mechanism because nomilin significantly decreased tert-butyl hydroperoxide-induced reactive oxygen species generation in the cells [65].

### 5.8. Anti-Osteoclastogenesis

Bone resorption occurs in the bone matrix and is mediated by mature osteoclasts [66]. In a study carried out by Kimira et al. [18], the effect of nomilin on the differentiation of mouse RAW 264.7 macrophage cells and mouse primary bone marrow-derived macrophages (BMMs) into mature osteoclasts was evaluated. The findings of this study indicated that treatment with nomilin significantly inhibited the formation of tartrate-resistant acid phosphatase (TRAP)-positive multinucleated osteoclasts from both RAW 264.7 cells and BMMs. Nomilin also decreased the bone-resorption activity and pit formation area of osteoclasts. Furthermore, nomilin could downregulate the expression levels of osteoclast genes, NFATc1 and TRAP, and suppress the receptor activator of NF-κB ligand (RANKL)-induced MAPK signaling pathways. Nomilin pretreatment delayed the progression of osteoarthritis in mice, suppressed the IL-1β induced over-regulation of pro-inflammation factors NO, IL-6, PGE2, iNOS, TNF-α, and COX-2, and down-regulated the degradation of the extracellular matrix induced by NF-κB/IL-1β signaling via disassociation of Keap1-Nrf2 in chondrocytes [67]. Together, the studies indicated that nomilin may be a potential therapeutic option in osteoarthritis.

### 5.9. Neuro-Protective function

A recent study [19] concluded that nomilin has a neuro-protective function on cerebral ischemia-reperfusion injury in rats by improvement in the infarct size, brain edema, and neurological deficits. Further, nomilin could attenuate blood-brain barrier (BBB) disruption and alleviate the loss of tight junction proteins, including ZO-1 and occluding-5. The findings of this study revealed that the protective effects of nomilin in cerebral ischemia-reperfusion rats were related to the nuclear factor erythroid 2-related factor 2 (Nrf2)/NAD(P)H dehydrogenase (quinone) 1 pathway.

## 6. The Pharmacological Properties of Nomilin Analogous

Obacunone and its glucosideis are highly oxygenated triterpenoid limonoid compounds rich in the Phellodendronchinense Schneid and Dictamnus dasycarpusb Turcz plant that are well-known for the anti-inflammatory properties in traditional medicine. Many studies show that obacunone may exert various pharmacological properties such as anti-oxidation, anti-inflammation and be used in the treatment for cancers, organ fibrosis, vascular diseases, neurodegenerative diseases, and metabolic diseases.

### 6.1. Obacunone and Cancers

It was reported that obacunone and obacunone glucoside may inhibit the proliferation, arrest the cell cycle at the G1 and G2/M phase, decrease the ratio of bcl2/bax mRNA, induce apoptosis through the induction of caspase-3, caspase-9 and p21, and cytochrome-c in the cytosol of SW480 colon cancer cells, indicating that both limonoids may induce apoptosis by activation of the intrinsic apoptosis pathway [68]. Similarly, obacunone and obacunone glucoside also suppress cell proliferation, and activate programmed cell death of prostate LNCaP cells through the down-regulation of Akt signaling and the inflammation-associated transcription factor, androgen receptor, and prostate-specific antigen [69]. In prostate Panc-28 cells, obacunone also exhibits the inhibitory effects on the cell proliferation, associated with phosphatidylserine translocation from the inner surface of the plasma membrane to the outer surface of the treated cells, and increases caspase-9 and caspase-3, upregulated expression of tumor suppressor protein p53, pro-apoptotic protein Bax, and downregulated anti-apoptotic protein Bcl2 [70]. A proteomics study showed that these anti-cancer effects of obacunone are closely related to the change of nicotinate and the nicotinamide metabolism, the phenylalanine metabolism, the tryptophan metabolism as well as the ascorbate metabolism in the tumor cells [71]. In an azoxymethane-induced tumorigenesis experiment, obacunone causes the reduction in the yield of aberrant crypt foci and the incidence of colonic adenocarcinoma. The ability to reduce the proliferating cell nuclear antigen-labeling index in crypts, and correlated with the prevention of aberrant crypt foci, indicates that obacunone possesses chemopreventive effects on chemically induced colon carcinogenesis, and obacunone might be useful for the prevention of human colon cancers [72,73].

Several studies have focused on the effects of obacunone on tumor-related inflammation. In azoxymethane and dextran sulfate sodium induced colorectal cancer mice, obacunone may inhibit the tumor size, bloody diarrhea, colon shortening, splenomegaly, and histological score. Meanwhile, obacunone suppresses colorectal cancer cell proliferation as well as the protein and mRNA levels of inflammatory cytokines in tumors [74]. In the breast cancer MCF-7 cell line, obacunone inhibits the proliferation, increases apoptosis by up-regulating the expression of the pro-apoptotic protein Bax and down-regulating Bcl2. The expression of inflammatory molecules including nuclear factor-kappa B (NF-κB) and cyclooxygenase-2 (COX-2) is also down-regulated through the activation of the p38 mitogen-activated protein kinase (MAPK) by obacunone treatment [75]. Obacunone attenuates pro-inflammatory mediators NO, IL-6, IL-1β, and MCP-1 at the transcriptional and translational levels. Mechanistically, obacunone may interact with macrophage migration inhibitory factors, and suppress p38-mediated AP-1 signaling by stabilizing the mRNA of mitogen-activated protein kinase phosphatase 1, and increase the expression time of the MKP-1 protein. Thus, obacunone may inhibit tumor growth partly through its anti-inflammatory effect [76].

Besides obacunone, several other analogues of nomilin such as nomilinic acid glucoside, deacetyl nomilinic acid glucoside, deacetyl nomilin, defuran nomilin, obacunone 17 beta d-glucopyranoside, nomilinic acid 17 beta d-glucopyranoside, and deacetylnomilinic acid 17 beta d-glucopyranoside also were identified to exhibit cytotoxicity and apoptosis induction effects on a variety of human cancer cell lines [3,10,13,77].

### 6.2. Obacunone and Digestive Diseases

A recent study showed that obacunone may alleviate intestinal inflammation and attenuate the symptoms in dextran sulfate sodium (DSS)-induced ulcerative colitis mice through the modulation of the gut microbiota composition and the attenuating toll-like receptor 4/NF-κB signaling. These effects of obacunone are probably related to the promotion of the expression of tight junction proteins TJP1 and occludin and suppression of inflammatory signaling cascades [78]. Obacunone also was shown to ameliorate liver fibrosis. In carbon tetrachloride-induced hepatic fibrosis mice, obacunone suppresses the TGF-β/Smad signals and epithelia mesenchymal transformations process, reduced levels of fibrosis markers α-SMA, collagen1, and vimentin, and exhibited an anti-oxidation effect by inhibiting reactive oxygen species via the activation of Nrf-2 and Glutathionperoxidase-4, a member of the glutathione peroxidase family alleviating oxidative stress, suggesting that obacunone could attenuate liver fibrosis via enhancing the Glutathionperoxidase-4 signal and inhibition of the TGF-β/Smad pathway [79].

### 6.3. Obacunone and Metabolic Disorders

In high glucose-induced oxidative damage in NRK-52E cells, obacunone protects the NRK-52E cells from the high glucose-induced decrease in cell viability and the accumulation of ROS. The protective effects of obacunone may be associated with an increase in the antioxidant levels such as SOD, GSH, and CAT, inhibit the production of ROS, and stabilize the mitochondrial membrane potential. Further study found that obacunone downregulated the activity of GSK-3β, enhanced the nuclear translocation of Nrf2 and its downstream genes NQO-1 and HO-1 in HG-treated cells, decreased the cytochrome c release from the mitochondria, indicating that obacunone may protect against high glucose-induced oxidative damage in the kidney via the inhibition oxidative stress and mitochondrial dysfunction [80]. Another study showed that obacunone lowered glycosylated hemoglobin, blood glucose, and white adipose tissue weight, and increased the weight of the gastrocnemius and quadriceps muscles in obese KKAy mice. Similar to nomilin, obacunone may stimulate TGR5, and inhibit adipocyte differentiation in 3T3-L1 cells via the antagonism of peroxisome proliferator-activated receptor γ transcriptional activity [81].

### 6.4. Obacunone and Vascular Diseases

Obacunone is also beneficial to the vascular disorders. A comparative study identified obacunone as an inhibitor of arginase I and II activities, which constrains endothelial nitric oxide synthase activity by substrate depletion and reduces nitric oxide bioavailability, in human umbilical vein endothelial cells. Obacunone also regulates vascular endothelial NO production, and increased nitrite/nitrate production, increased intracellular l-arginine concentration and enhanced eNOS coupling in isolated aortic rings, resulted in the increased NO and decreased superoxide production, suggesting that obacunone may treat cardiovascular diseases resulting from endothelial dysfunction [82].

### 6.5. Obacunone and Wound Healing

Obacunone also was reported to increase wound healing, and migration rates and osteoblast differentiation. Obacunone increased the mRNAs levels of the bone differentiation markers, Alp, bone sialoprotein, osteopontin, osteocalcin, and bone morphogenetic protein, and the phosphorylation of smad1/5/8. This effect may occur through the enhancement of the expression of runt-related transcription factor 2 and the protein level of β-catenin, and inhibition of GSK3β during osteoblast differentiation. The study demonstrated that obacunone may activate osteoblast differentiation and bone mineralization, suggesting that obacunone may be a therapeutic selection for bone diseases such as osteoporosis and periodontitis [83].

### 6.6. Obacunone and Neurodegenerative Diseases

Obacunone exerts the neuroprotective effects on glutamate-induced oxidative injury neurotoxicity in the mouse hippocampal HT22 cells. Meanwhile, obacunone increased p38 MAPK phosphorylation and induced HO-1 expression via the p38 MAPK pathway, suggesting that obacunone could be effective candidates for the treatment of oxidation-related neuronal degeneration diseases such as Parkinson’s disease and Alzheimer’s disease [84]. In primary murine retinal pigment epithelium ARPE-19 cells, obacunone inhibited UVR-induced ROS accumulation, mitochondrial depolarization, lipid peroxidation and single strand DNA accumulation, alleviated UVR-induced retinal pigment epithelium cell death and apoptosis, activated Nrf2, mediated the expression of antioxidant genes. In mice, obacunone inhibited light-induced retinal damage, indicating that obacunone protects ultra-violet radiation induced oxidative injury in retinal pigment epithelium cells through activation of the Nrf2 signaling cascade [85].

### 6.7. Anti-Bacterial Effects of Obacunone

Obacunone was found to interfere with autoinducer-mediated cell-cell signaling in Vibrio harveyi and Escherichia coli O157:H7 to repress the locus of enterocyte effacement [86]. Obacunone also inhibits the food-borne pathogen Salmonella enterica serovar Typhimurium LT2 through the repress of Salmonella pathogenicity island 1 (SPI1) and SPI2, the maltose transporter, and the hydrogenase operon. The repression of SPI1 probably is mediated through the inhibition of hilA. These findings suggest that obacunone may possess antivirulence activity on some bacteria and may be a lead compound for development for anti-bacterium agents [87].

## 7. Conclusions

The content of nomilin in citrus fruits depends on various factors, including the variety of the fruit, the season of harvest, etc. In recent decades, a considerable number of experimental studies were carried out to investigate the health benefits of nomilin and its analogues. It is possible that an adequate intake of citrus fruit may activate phase II enzymes in the body, which have potential effects on the prevention of cancer and other xenobiotic-related diseases. However, the oral bioavailability of nomilin is low, which results from its poor solubility in the aqueous phase of the digestive fluids, and its rapid hepatic metabolism. Thus, further studies should focus on exploring methods which can improve the bioavailability of nomilin and better investigate the effects of nomilin in humans.

## Figures and Tables

**Figure 1 molecules-28-00269-f001:**
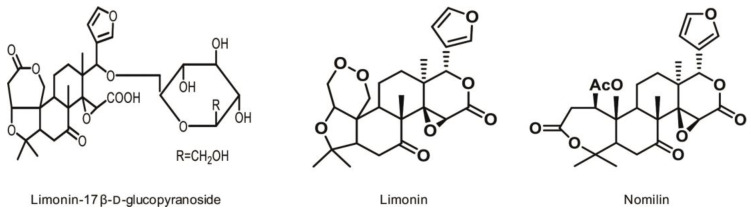
The most abundant limonoid glucoside and aglycones.

**Figure 2 molecules-28-00269-f002:**
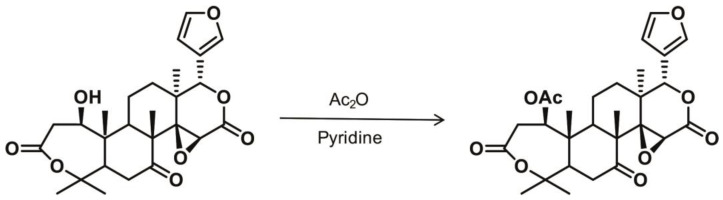
Chemical synthesis of nomilin.

**Figure 3 molecules-28-00269-f003:**
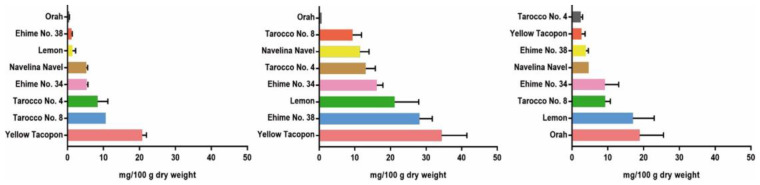
Nomilin content of citrus varieties at different fruit growth stages (mean ± SD). Left: falling stage; middle: young fruit stage; right: mature stage.

## Data Availability

Not applicable.

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
