# Peer review of "Nomilin and Its Analogues in Citrus Fruits: A Review of Its Health Promotion Effects and Potential Application in Medicine"

_molecules, 2022, doi:10.3390/molecules28010269_

Round 1

Reviewer 1 Report

I read with interest the study by Zhou and colleagues, evaluating “Nomilin and its analogues in citrus fruits: A review of its health promotion effects and potential application in medicine”.

This article chronicled the evolution of nomilin research from examining its history, structure, occurrence, multiple biological activities and disease prevention properties, and the pharmacological properties of nomilin analogues represented by obacunone. Although this study provides detailed information to help us gain an in-depth understanding of nomilin, there are still some problems in the manuscript, please consider my comments (below) and revise the manuscript to enhance its quality.

1. Page 1, line 30. A spacing symbol is missing between "17" and "β", and the same problem occurs in Figure 1.

2. Page 2, line 51. "but the exact place of origin of citrus fruits is still under controversy." This sentence is transitive and consequential. It would be more appropriate to put it after "It is considered..." to achieve the purpose of better transition to the discussion of the origin.

3. Page 2, line 70. "in 1947" is missing a period.

4. Page 2, line 73. The title number is incorrect, it should be the third point.

5. Page 2, line 77. Delete "see" to unify the expression.

6. Page 4, line 151. Please provide more detailed description and data for the nanoparticle-based delivery system.

7. Page 4, line 171. It should be marked out which stage of growth or maturation each graph corresponds to.

8. Page 5, line 211. Why does the antioxidant of nomilin equal to that of cinnamic acid indicate that it does not possess inherent antioxidant capacity? Is there any other study that also think that nomilin cannot be used as an antioxidant?

9. Page 10, line 463. The botanical origin of nomilin has been described several times in the Abstract and Introduction, so please avoid repeating it over and over again. A clear and concise summary of the key points needs to be provided in conjunction with the review. It is recommended that this section be rewritten.

10. Page 11, line 484. There are missing page numbers and formatting problems in the references, please check and correct them carefully.

Author Response

We thank the reviewer for the positive comments and helpful suggestion. We have revised the manuscript according to the suggestion.

  1. Page 1, line 30. A spacing symbol is missing between "17" and "β", and the same problem occurs in Figure 1.

Response: We added spacing symbol between "17" and "β" in the text and Figure legends.

  1. Page 2, line 51. "but the exact place of origin of citrus fruits is still under controversy." This sentence is transitive and consequential. It would be more appropriate to put it after "It is considered..." to achieve the purpose of better transition to the discussion of the origin.

Response: We are grateful to the reviewer for helpful suggestion. Now we have revised the sentence accordingly.

  1. Page 2, line 70. "in 1947" is missing a period.

Response: We changed "in 1947" to "in 1948".

  1. Page 2, line 73. The title number is incorrect, it should be the third point.

Response: We corrected "2" as "3".

  1. Page 2, line 77. Delete "see" to unify the expression.

Response: We deleted “see”.

  1. Page 4, line 151. Please provide more detailed description and data for the nanoparticle-based delivery system.

Response: We have added more information about the nanoparticle-based delivery system.

  1. Page 4, line 171. It should be marked out which stage of growth or maturation each graph corresponds to.

Response: We added “Left: falling stage; middle: young fruit stage; right: mature stage” to the Figure legends.

  1. Page 5, line 211. Why does the antioxidant of nomilin equal to that of cinnamic acid indicate that it does not possess inherent antioxidant capacity? Is there any other study that also think that nomilin cannot be used as an antioxidant?

Response: In the cited reference, cinnamic acid was used as a negative control for the anti-oxidant activity, and the data indicate that nomilin does not possess antioxidant capacity. We searched the documents, and did not find other study showing negative antioxidant effects of nomilin.

  1. Page 10, line 463. The botanical origin of nomilin has been described several times in the Abstractand Introduction, so please avoid repeating it over and over again. A clear and concise summary of the key points needs to be provided in conjunction with the review. It is recommended that this section be rewritten.

Response: We fully agree to the comments and deleted the sentence.

  1. Page 11, line 484. There are missing page numbers and formatting problems in the references, please check and correct them carefully.

Response: We added the missing information to the references.

Reviewer 2 Report

An interesting review about the potential application in medicine  of the nomilin and its analogues in citrus fruit.  A complete analysis of the properties of nomilin was made including detossification, anti-oxidant, anti-cancer, ecc. 

The analysis of the pharmacological properties of nomilin analogues was also very interesting. As it was said in the reiew more studies need to be done as the oral bioavailability of nomilin is low, which results from its poor solubility in the aqueous phase of digestive fluids, and its rapid hepatic metabolism. Furthermore the references are appropriated.

Author Response

An interesting review about the potential application in medicine  of the nomilin and its analogues in citrus fruit.  A complete analysis of the properties of nomilin was made including detossification, anti-oxidant, anti-cancer, ecc. 

The analysis of the pharmacological properties of nomilin analogues was also very interesting. As it was said in the reiew more studies need to be done as the oral bioavailability of nomilin is low, which results from its poor solubility in the aqueous phase of digestive fluids, and its rapid hepatic metabolism. Furthermore the references are appropriated.

Response: We thank the reviewer for helpful comments and suggestion. We tried to search more reference, but could not find recent publication regarding the metabolism of nomilin.

Reviewer 3 Report

The structure of Figure 2 should be revised.

Author Response

The structure of Figure 2 should be revised.

Response: We thank the reviewer for pointing this out. Now we revised Figure 2 accordingly.

Round 2

Reviewer 1 Report

No more questions.